# Implication of Hepsin from Primary Tumor in the Prognosis of Colorectal Cancer Patients

**DOI:** 10.3390/cancers14133106

**Published:** 2022-06-24

**Authors:** David Zaragoza-Huesca, Andrés Nieto-Olivares, Francisco García-Molina, Guillermo Ricote, Sofía Montenegro, Manuel Sánchez-Cánovas, Pedro Garrido-Rodríguez, Julia Peñas-Martínez, Vicente Vicente, Francisco Martínez, María Luisa Lozano, Alberto Carmona-Bayonas, Irene Martínez-Martínez

**Affiliations:** 1Centro Regional de Hemodonación, Department of Haematology and Medical Oncology, Hospital General Universitario Morales Meseguer, University of Murcia, IMIB-Arrixaca, 30100 Murcia, Spain; davidzaragozahuesca5369@gmail.com (D.Z.-H.); guille000000@hotmail.com (G.R.); sofiamontenegroluis@gmail.com (S.M.); manuelsanchezcanovas@gmail.com (M.S.-C.); pedro.garridor@outlook.es (P.G.-R.); jpenas@um.es (J.P.-M.); vicente.vicente@carm.es (V.V.); mllozano@um.es (M.L.L.); 2Department of Pathology, Hospital General Universitario Morales Meseguer, 30008 Murcia, Spain; andres.nieto2@carm.es; 3Department of Pathology, Hospital General Universitario Reina Sofía, 30003 Murcia, Spain; pacogm@um.es (F.G.-M.); fmdiaz@um.es (F.M.); 4Centro de Investigación Biomédica en Red de Enfermedades Raras, U-765-CIBERER, Instituto de Salud Carlos III, 28220 Madrid, Spain

**Keywords:** hepsin, colorectal cancer, thrombosis, metastasis, hepsin paradox

## Abstract

**Simple Summary:**

Hepsin is a serine protease whose deregulation leads to tumor invasion and metastasis in many tumor types. In colorectal cancer, the role of hepsin is unknown, so we aimed to study the correlations between its expression and clinical-histopathological variables of patients with this cancer. We recruited 169 patients with localized disease and 118 with metastatic cancer at diagnosis, and then, we measured hepsin staining from primary biopsy in order to correlate it with time-to-event variables, laboratory data, genetic alterations, histologic features, etc. Our results demonstrated hepsin was an independent prognosis factor for metastasis and thrombosis in patients with localized disease, whereas among metastatic subjects, lower levels of hepsin were associated with highest tumor dedifferentiation and spread to distant organs. These results point to hepsin as a potential biomarker for considerable complications in patients with colorectal cancer. In addition, this article brings to light new information about the implication of hepsin in colorectal cancer.

**Abstract:**

Hepsin is a type II transmembrane serine protease whose deregulation promotes tumor invasion by proteolysis of the pericellular components. In colorectal cancer, the implication of hepsin is unknown. Consequently, we aimed to study the correlations between hepsin expression and different clinical-histopathological variables in 169 patients with localized colorectal cancer and 118 with metastases. Tissue microarrays were produced from samples at diagnosis of primary tumors and stained with an anti-hepsin antibody. Hepsin expression was correlated with clinical-histopathological variables by using the chi-square and Kruskal–Wallis tests, Kaplan–Meier and Aalen–Johansen estimators, and Cox and Fine and Gray multivariate models. In localized cancer patients, high-intensity hepsin staining was associated with reduced 5-year disease-free survival (*p*-value = 0.16). Medium and high intensity of hepsin expression versus low expression was associated with an increased risk of metastatic relapse (hazard ratio 2.83, *p*-value = 0.035 and hazard ratio 3.30, *p*-value = 0.012, respectively), being a better prognostic factor than classic histological variables. Additionally, in patients with localized tumor, 5-year thrombosis cumulative incidence increased with the increment of hepsin expression (*p*-value = 0.038). Medium and high intensities of hepsin with respect to low intensity were associated with an increase in thrombotic risk (hazard ratio 7.71, *p*-value = 0.043 and hazard ratio 9.02, *p*-value = 0.028, respectively). This relationship was independent of previous tumor relapse (*p*-value = 0.036). Among metastatic patients, low hepsin expression was associated with a low degree of tumor differentiation (*p*-value < 0.001) and with major metastatic dissemination (*p*-value = 0.023). Hepsin is a potential thrombotic and metastatic biomarker in patients with localized colorectal cancer. In metastatic patients, hepsin behaves in a paradoxical way with respect to differentiation and invasion processes.

## 1. Introduction

Hepsin (HPN) is a type II transmembrane serine protease (TTSP), a family of proteins characterized by a short intracellular amino-terminal domain, a hydrophobic transmembrane domain and an extracellular carboxyl-terminal catalytic domain whose active center is formed by the catalytic triad serine, aspartic acid and histidine [1]. HPN is mainly expressed in the liver, although it also exerts its function in other tissues and organs, such as adipose tissue, the kidney or the inner ear [2]. Among its functions are the regulation of cell growth [3], its participation in the hepatic metabolism of glycogen and lipids [4], the degradation of the extracellular matrix [5] and the activation of procoagulant factors, such as factor VII [6,7], which leads us to think of its potential capacity to initiate the coagulation cascade.

In cancer, many extracellular proteases present deregulated levels, which contribute to tumor progression through cleavage of extracellular matrix components [8]. Although HPN is part of this proteolytic complex, contributing to a degradation of the pericellular microenvironment, its net effect on tumorigenesis is complex and context dependent. Thus, in prostate cancer, HPN has a clear pro-tumorigenic function by promoting invasion processes in certain settings [9]. However, other reports in the literature suggest that very high levels of this serine protease may exert paradoxical antitumor effects with limitation of oncogenic signaling and increased autophagy [10]. This complex effect explains some discrepant observations reported in the literature. Thus, in ovarian cancer, HPN contributes to tumor progression [11], whereas in endometrial cancer, it inhibits tumor cell growth [12]. In gastric cancer, elevated HPN levels are associated with a worse prognosis [13], whereas in breast cancer, low HPN levels predict worse survival [14]. In addition to these opposing effects, the so-called “HPN paradox” has been described as a dynamic phenomenon, whereby, after playing an active role in early tumorigenesis, HPN expression is reduced in more advanced stages to preserve cellular fitness, leading to suppression of its expression in metastatic cells (i.e.**,** limitation of proteolytic excesses once the invasive process is complete) [10].

A limited number of reports in the literature would suggest that HPN might play a more or less crucial role in colorectal cancer (CRC), although the relevance of the enzyme in this context has not been thoroughly evaluated. First, cell invasion processes based on proteolytic systems have been extensively studied and are relevant in CRC from early transformation to advanced tumorigenesis [15]. The individual contribution of HPN in the network of proteases operating in CRC is unknown, although previous work points to increased serum levels of HPN in patients with advanced versus localized CRC, especially during treatment with chemotherapy [16]. Beyond this, the role of HPN along the CRC continuum is unknown, including the association of the protease with relevant clinical events or with thrombotic manifestations [17]. Based on the biological actions of the enzyme, we wanted to investigate whether HPN could play a key role in CRC. Specifically, in this retrospective study, we evaluated the presence of HPN in CRC samples, representative of the evolutionary spectrum of this tumor, to assess the association of its expression with clinicopathological features, survival endpoints and thrombotic risk.

## 2. Materials and Methods

### 2.1. Patients

Patients with localized or metastatic CRC at diagnosis were included in this study. Cases with localized CRC who eventually developed metastases during follow-up were only analyzed within the group of localized neoplasms. The eligibility criteria for patients with localized tumors were: histological diagnosis of colorectal adenocarcinoma, tumors with TNM stage II–III and microscopic resection with negative margins (R0). The eligibility criteria for metastatic neoplasms were: presence of metastatic disease at diagnosis and use of at least one line of chemotherapy for advanced disease. Of the overall cohort of 287 patients with CRC specimens at diagnosis, 169 had localized disease, and 118 had metastatic disease.

Patients were recruited from two centers (Hospital General Universitario Morales Meseguer (HMM) and Hospital General Universitario Reina Sofía de Murcia), with equivalent diagnostic protocols and with clinical management in a single oncology service, similar reference population and shared coloproctology committees. Consecutive recruitment was requested in both hospitals.

The project was submitted to and approved by the Clinical Research Ethics Committee of the HMM. The study was conducted in accordance with the Biomedical Research Law 14/2007. Data confidentiality was guaranteed at all times, in accordance with the Organic Law 15/1999 on personal data protection, including the rights of access, rectification, cancellation and opposition of data. Informed consent was requested from all patients for their inclusion in the study.

### 2.2. Tissue Microarrays and Immunohistochemistry

Once the primary tumor samples were obtained, on which the diagnosis was made, two pathologists selected the areas with the highest density of tumor cells, and 2 mm diameter cylinders were extracted. Tissue microarrays (TMAs) were constructed from these cylinders with UNITMA equipment (Quick-Ray, Manual Tissue Microarrayer (UNITMA, Co., Ltd**.**, Seoul, South Korea)). Immunohistochemistry of TMAs was performed with a specific anti-HPN antibody (Anti-HPN; HPA006804-100UL; Sigma-Aldrich, Madrid, Spain) using Autostainer Link 48-DAKO automated systems. Each TMA contained patient samples in duplicate; healthy tissue samples from the stomach and colon were used as positive controls for HPN staining, and spleen was used as a negative control.

The two pathologists independently evaluated the intensity of HPN immunohistochemical staining semi-quantitatively at three possible intensities: high, medium or low. For reference, controls with positive and negative HPN staining were used, as well as THE HUMAN PROTEIN ATLAS open-access database [18]. This database has a library of primary tissue biopsies from different cancer types (including CRC) in which HPN immunohistochemical staining is measured according to these three levels of intensity [18].

### 2.3. Clinical and Histopathological Variables of the Patients

The clinical variables evaluated were disease-free survival (DFS), progression-free survival (PFS), overall survival (OS) and cumulative incidence of thrombosis. In patients with localized tumor, DFS was defined as the time interval between localized tumor surgery and metastatic relapse. In patients with advanced tumor, PFS was defined as the interval between initiation of first-line chemotherapy and tumor progression. In both groups of patients, the OS and cumulative incidence of thrombosis were defined from the date of disease diagnosis. In all cases, subjects with no events at the last follow-up were censored. Thrombotic disease was diagnosed using imaging techniques (Doppler ultrasound or computed tomography), according to clinical practice.

To model the association between time-to-event endpoints and HPN staining, multivariate models were applied. Model building was performed by understanding the causal mechanisms or sources of bias in CRC. Thus, the covariates for the multivariate model were chosen on theoretical grounds, following a literature review [19,20,21,22,23,24], and taking into account expert opinion. In metastatic tumors, the multivariable model for PFS and OS included HPN staining (specified as a three-level categorical variable), Eastern Cooperative Oncology Group (ECOG) performance status, histologic grade, presence of more than one metastatic site and specific metastatic sites (peritoneal, lung and liver) at diagnosis. In localized tumors, the multivariable model for DFS included HPN staining, histologic grade, lymphovascular invasion and TNM stage (II versus III) at diagnosis. In both tumor groups, the multivariable model for the cumulative incidence of thrombosis included the covariates histologic grade and lymphovascular invasion at diagnosis. In addition, for localized tumors, it included TNM stage (II versus III) at diagnosis, and for metastatic tumors, the presence of more than one metastatic focus and specific metastatic foci at diagnosis. Other factors of interest were demographic variables, laboratory data (carcinoembryonic antigen (CEA), lactate dehydrogenase, leukocytes and blood platelets), genetic alterations (KRAS/NRAS/BRAF mutations and microsatellite instability) and other histopathologic variables at diagnosis: perineural invasion, presence and number of tumor deposits distant from the primary tumor, size and extent of the primary tumor (T-stage) and extent of tumor that had spread to nearby lymph nodes (N-stage).

### 2.4. Statistics

The study had a fixed sample size, limited by the availability of tissue samples present in the hospitals where patients were recruited. Therefore, inference should be interpreted according to the width of the confidence interval (CI). Statistical analysis between HPN immunohistochemical staining and clinicopathologic features was performed in the overall cohort and stratified for localized and metastatic tumors. Correlation between discrete variables and HPN was performed using the chi-square tests [25]. In the case of continuous variables, the Kruskal–Wallis tests were used [25]. Time-to-event outcomes were evaluated using the Kaplan–Meier estimator and Log-rank test for trends, and with the Cox proportional hazards multivariate regression [26,27]. For the cumulative incidence of thrombosis, Gray**’**s test [28] and the multivariate Fine and Gray test [29] were also used. For the calculation of such incidence, the Aalen–Johansen estimator [30] was also used, taking into account relapse as a competing event [31], to discern whether or not these thromboses were harbingers of relapse [17]. All analyses were performed using R-4.1.2 statistical software [25], including the survival [32], and cmprsk [33] packages.

## 3. Results

### 3.1. Patients and Clinical-Histopathologic Data

The clinicopathologic characteristics of the overall cohort, localized patients and metastatic patients are shown in Table 1. In patients with localized disease, the majority of tumors were T-stage 3 (62.7%, number of patients (N) = 106/169), followed by T-stage 4 (12.7%, N = 21/169). Nodal disease (N-stage > 0) was present in 53.3% of the cases (N = 90/169). In patients with advanced disease, the most frequent metastatic location at diagnosis was the liver (71.2% (N = 84/118)), and the involvement of at least two distant organs occurred in 39% (N = 46/118) of patients. The remaining characteristics are in accordance with those expected in CRC cohorts treated in clinical practice (Table 1). The median follow-up of patients with localized disease was 41.2 months (range 3.8–190.8) compared to 24.5 months (range 1.4–194) for those with metastatic disease. In localized tumors, 54/169 (32%) patients had a metastatic relapse during follow-up. The median OS in patients with localized tumors was 93 months (95% CI, 83.2–118.2). At the time of analysis, the median OS of subjects with metastatic tumors was 24.7 months (95% CI, 21.5–33.5), and 102/118 (86.4%) had progressed. In patients with localized or metastatic cancer, the cumulative incidence of thrombosis at 5 years was 18.24% (95% CI, 12.69–25.84) and 39.39% (95% CI, 29.05–51.85), respectively (Table 1).

Appendix A show the relationship of tumor HPN expression with discrete, time-to-event and continuous variables, respectively.

### 3.2. Low HPN Staining Intensity Predisposes to Poor CRC Differentiation

Analysis of the relationship between HPN expression and histological grade of tumor differentiation reflects that globally, there is an increase in HPN expression from well-differentiated to moderately differentiated tumors, which is particularly evident in the case of advanced neoplasms (Figure 1). In the latter, the high intensity of HPN expression ranged from 27% in well-differentiated tumors to 78.1% in moderately differentiated tumors (chi-square = 31.4, degrees of freedom (df) = 4, *p*-value < 0.001) (Appendix A). This trend, however, was reversed in the transition from moderately differentiated to poorly differentiated (Figure 1). Furthermore, cases of tumors with low HPN staining intensity increased from those with moderate degree of differentiation to poorly differentiated (localized: 18.9% versus 50%, chi-square = 9.15, df = 4, *p*-value = 0.056; metastatic: 12.5% versus 44.4%, chi-square = 31.4, df = 4, *p*-value < 0.001) (Appendix A).

### 3.3. In Patients with Localized Disease, High-Intensity HPN Staining Decreases DFS

As shown in Figure 2, in these patients, HPN staining intensity was associated with worse 5-year DFS rate: 51% (95% CI, 37–71%), 59% (95% CI, 46–75%) and 73% (95% CI, 56–96%) for tumors with high, medium and low staining, respectively (Appendix A). In the multivariable Cox regression, medium and high HPN expressions were related to poor prognosis, with hazard ratios (HRs) for recurrence of 2.83 (95% CI, 1.07–7.48; *p*-value = 0.035) and 3.30 (95% CI, 1.29-8.40; *p*-value = 0.012), respectively, versus low expression (Table 2). The Cox multivariable model reflected that HPN staining provided additional prognostic information to the classical histopathological factors (histological grade, lymphovascular invasion and TNM stage) (Table 2).

### 3.4. The Intensity of HPN Staining in Patients with Localized Disease at Diagnosis was Related to the Cumulative Incidence of Thrombosis

At the time of analysis, 65/287 (22.6%) thrombosis events were recorded, of which 30/169 (17.8%) occurred in patients with localized disease and 35/118 (29.7%) in metastatic ones. The locations of thrombosis in each cohort are detailed in Table 3. In patients with localized disease, the 5-year cumulative incidence of thrombosis in tumors with high, medium and low HPN staining was 23% (95% CI, 12–33%), 22% (95% CI, 9–33%) and 0%, respectively (Log-rank for trends *p*-value = 0.038 and Gray’s test *p*-value = 0.009; Appendix A and Figure 3). In the multivariable Fine and Gray model in patients with localized disease, medium and high HPN staining increased the cumulative incidence of thrombosis, with HR of 7.705 (95% CI, 1.06–55.92; *p*-value = 0.043) and 9.016 (95% CI, 1.27–63.80; *p*-value = 0.028), respectively, versus low staining (Table 4). As in the case of DFS, HPN expression conferred a better prognostic value for thrombosis risk than the other histopathological variables used in the model (Table 4). Appendix A shows the detailed and individualized description of the circumstances under which thrombosis occurred in patients with localized disease, in relation to other possible factors that could modulate thrombotic risk other than HPN itself. In patients with localized disease, excluding those who relapsed before the vascular event, the 5-year incidence of thrombotic complication when HPN staining intensity was high, medium and low was 16.3% (95% CI, 7.4–25.1%), 7.8% (95% CI, 1.2–14.4%) and 0% (95% CI, 0–0%), respectively (Figure 4). The *p*-value of the Aalen–Johansen estimator was significant (*p*-value = 0.036).

### 3.5. Association of HPN with the Prognosis of Metastatic Patients and Their Cumulative Incidence of Thrombosis

In the cohort of metastatic patients at diagnosis, at the time of analysis, there were 102/118 (86.4%) progression events after first-line chemotherapy and 95/118 (80.5%) deaths. HPN staining in the primary tumor was not substantially associated with PFS or OS nor with thrombotic risk (Appendix A). In contrast, according to the multivariable Cox model, other variables, such as poor tumor differentiation, were related to worse prognosis, with HR of 2.91 for PFS (95% CI, 1.62–5.25; *p*-value < 0.001) and 3.59 for OS (95% CI, 1.38–6.33; *p*-value < 0.001) (Appendix A). Similarly, poorly differentiated histological grade was associated with an increased risk of thrombosis, with a HR of 2.85 (95% CI, 1.11–7.33; *p*-value = 0.029) (Appendix A).

Since the RAS oncogene has been described as promoting tumor growth and invasion through HPN activation [34], we wanted to determine whether among the metastatic patients with mutated RAS there was a correlation between HPN and the time-to-event variables mentioned above. The results reflected that among metastatic patients with RAS gene mutations, HPN staining in the primary tumor was not substantially associated with PFS, OS or cumulative incidence of thrombosis, unlike histopathological factors, which did show that correlation, such as histological grade, some metastatic locations or the ECOG scale (Appendix A).

### 3.6. Association of HPN with the Pattern of Tumor Dissemination in Metastatic Patients

Figure 5 shows, for each type of metastatic involvement at diagnosis, the percentage of patients in the different HPN staining intensity groups. HPN staining was not substantially different between patients with or without liver metastases (Figure 5A and Appendix A). For peritoneal metastases, cases with low HPN staining intensity were higher among patients with such involvement (34.4%) than without it (18.6%), although the differences were not significant (*p*-value = 0.11) (Figure 5C and Appendix A). Among cases with lung metastasis, high HPN staining decreased its percentage with respect to no lung involvement (32.3% versus 47.1%), in the opposite direction to low staining (32.3% versus 19.5%), although these differences were also not significant (*p*-value = 0.245) (Figure 5B and Appendix A). Significantly, the percentage of patients with high intensity of HPN staining was lower in those with >1 metastatic foci at diagnosis compared to those with a single metastatic focus (30.4 versus 51.4%, respectively; *p*-value = 0.023). The distribution of patients with low-intensity HPN staining was reversed (34.8 versus 15.3%; *p*-value = 0.023) in patients with >1 versus 1 metastasis (Figure 5D and Appendix A).

## 4. Discussion

Proteolytic enzymes are involved in the destruction of the extracellular matrix of CRC, forming a sophisticated network that participates in the processes of invasion and metastasis. Therefore, these enzymes could be postulated both as potential biomarkers and as possible therapeutic targets [15]. In this study, we evaluated the influence of the expression of HPN, a transmembrane serine protease, on the prognosis of localized and metastatic CRC. Our results support the hypothesis that a high expression of HPN in the tumor triples the risk of recurrence after surgery for localized CRC, being a better prognostic factor than other classic histopathological factors [35]. Furthermore, in patients with localized disease, increased expression of HPN compared to patients with low staining raises thrombotic risk, independently of tumor recurrence. In contrast, our results do not support the notion that HPN staining in primary tumor samples influences survival or thrombotic risk in patients with metastatic CRC. In the latter, the paradox of HPN [10] is reflected in the association of lower expression of this serine protease with a lower degree of histological tumor differentiation and greater metastatic spread to distant organs.

CRC is one of the most common tumors and has a variable prognosis depending on the stage, with OS at 5 years being approximately 90% for patients with stage I and slightly more than 10% for patients with stage IV [36,37]. These data reflect that early diagnosis of CRC and the prediction and prevention of metastasis can be decisive for patient survival. Furthermore, CRC, like other tumors of the gastrointestinal tract, is associated with a high risk of thrombotic events, such as pulmonary thromboembolism [38]. Thrombosis correlates with increased morbidity and mortality. In fact, some thrombotic markers are associated with a worse prognosis in patients with CRC, such as platelet count [39], D-dimer levels [40], fibrinogen [41] and the von Willebrand factor [42].

Therefore, the identification of prognostic markers of metastasis or thrombosis in CRC could help anticipate the appearance of these complications, thus improving the survival of patients. In this context, we highlight that in the cohort of patients with localized disease, the progressive increase in the intensity of HPN staining was accompanied by a lower DFS. This is in agreement with other studies on HPN in prostate and breast cancer, and the underlying functional mechanism could be based on the ability of HPN to promote invasion to distant organs through the disorganization of the basement membrane that surrounds the primary tumor [43,44,45,46]. Significantly, according to Cox’s multivariate model for DFS, HPN was an indicator of relapse risk, and its prognostic value was much more accurate than other histopathological covariates, such as histological grade, lymphovascular invasion and TNM stage. Thus, given that there are articles that discredit the usefulness of histopathological variables to predict events related to DFS in CRC [47], we propose HPN as a potential biomarker to predict metastatic relapse.

Another interesting finding among our results in the cohort of patients with localized disease is that the increase in HPN staining in the primary tumor significantly increased the cumulative incidence of thrombosis, with HPN again presenting a better prognostic value than the histopathological covariates. In addition, this relationship was maintained when metastatic recurrence was excluded as a possible underlying cause of thrombosis. The interaction between HPN and coagulation in this type of cancer could be explained by the ability of HPN to activate proteins, such as factor VII, XII and IX of the coagulation cascade [6,7,48], although none of these studies have been conducted in humans. Another key fact to explain the association between HPN and thrombosis is that, although this protein is described as a transmembrane protease, there are studies that have identified it in the serum of patients with CRC [16]. Therefore, from the primary tumor, the cells could release the extracellular fraction of HPN into the microvasculature that irrigates the tumor, accessing the bloodstream, where its interaction with coagulation factors would be facilitated.

The effect of HPN on relapse and thrombosis is particularly interesting because, despite tumor resection [49], high levels of HPN at diagnosis maintain an effect on the occurrence of thrombosis and metastatic relapse during follow-up. We do not know the mechanism by which HPN maintains this effect when the tumor is removed, and future studies will be necessary to understand this association. However, a possible explanation could be based on the ability of HPN to degrade the surrounding extracellular matrix [5] and promote tumor cell motility and invasion [50] in areas of healthy cells, which could make total tumor resection difficult. Another possible explanation for the prothrombotic effect is based on the previous arguments about the ability of HPN to enter the circulation [16], so there would be a postoperative soluble fraction that could promote a state of hypercoagulability.

In patients with metastatic CRC at diagnosis, the relationship of HPN with tumor histological grade and metastatic spread supported “the HPN paradox”, according to which tumors have developed a precise spatiotemporal restriction of HPN overexpression [10]. Therefore, our results showed that high expression of HPN contributed to achieving moderate tumor differentiation. However, HPN expression decreased in poorly differentiated tumors, with patients with lower HPN staining being the most abundant in this histological grade group. In addition, we found that in those subjects with two or more organs affected by metastases, the expression of HPN was significantly reduced with respect to patients with less metastatic dissemination. In summary, our data argue that as CRC becomes more differentiated from healthy tissue and expands its range of invaded distant organs, HPN expression levels decrease, possibly because this serine protease is no longer required. An example of a utility-dependent tumor expression of HPN occurs in prostate cancer, where this “HPN paradox” was described [51]. In the latter, while primary tumors increased HPN levels to promote tumor progression, cells that reached distant tissues and gave rise to metastases reduced their HPN levels. In this case, HPN would no longer be necessary in an environment in which the tumor cell seeks to adhere to the new surrounding matrix to form a new tumor niche, instead of degrading it [51].

When comparing patients with metastatic and localized disease, we found no differences in the levels of HPN staining (Appendix A), but it should be noted that only in localized neoplasms did we find prognostic value for HPN. We do not know why the predictive effect of HPN on thrombosis and metastasis disappears in patients with metastatic disease. We can hypothesize that in primary tumors that have already metastasized, HPN would lose much of its proinvasive utility and, consequently, its relationship with tumor progression would also be lost. In addition, for both metastatic relapse and thrombosis, our analysis does not include the potential presence of HPN in plasma [16] or metastatic locations [52]. In this sense, in patients with metastatic cancer, by considering only the expression of HPN in the primary tumor, we could be underestimating the total levels of HPN in the time-to-event analysis.

Among the limitations of this study, the first is the unique origin of the recruited patients, since they all belong to the same geographical area (Murcia Region, Spain), so the extrapolation of these results to larger cohorts from other distant geographical areas must be confirmed. The second limitation refers to the results regarding HPN as a thrombotic biomarker. Although the differences in the cumulative incidence of thrombosis are significant according to the expression of this protein in the primary tumor biopsy, thrombosis, which very often accompanies cancer, has a multifactorial etiology that involves not only genetic factors but also numerous environmental factors that could interfere with the usefulness of HPN as a biomarker.

## 5. Conclusions

In conclusion, our article describes HPN as a prognostic marker of metastatic recurrence and thrombosis in patients with localized CRC. If validated in an independent cohort, the ability of HPN to predict metastatic relapse or thrombosis could contribute to the prevention of these complications, which could be decisive for the survival of patients with localized CRC. In addition, in metastatic cancer, HPN expression in the primary tumor appears to be subjected to paradoxical regulation in the processes of tumor differentiation and invasion of distant organs.

## Figures and Tables

**Figure 1 cancers-14-03106-f001:**
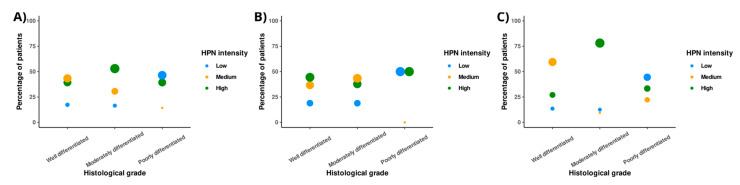
Distribution of Hepsin staining intensity in patients with different grades of tumor differentiation. (**A**) Percentage of patients in the overall cohort with different Hepsin staining intensity in each of the three grades of tumor histological differentiation. (**B**) Percentage of localized patients with different Hepsin staining intensity in each of the three grades of tumor histological differentiation. (**C**) Percentage of metastatic patients with different Hepsin staining intensity in each of the three grades of histological tumor differentiation. HPN: Hepsin.

**Figure 2 cancers-14-03106-f002:**
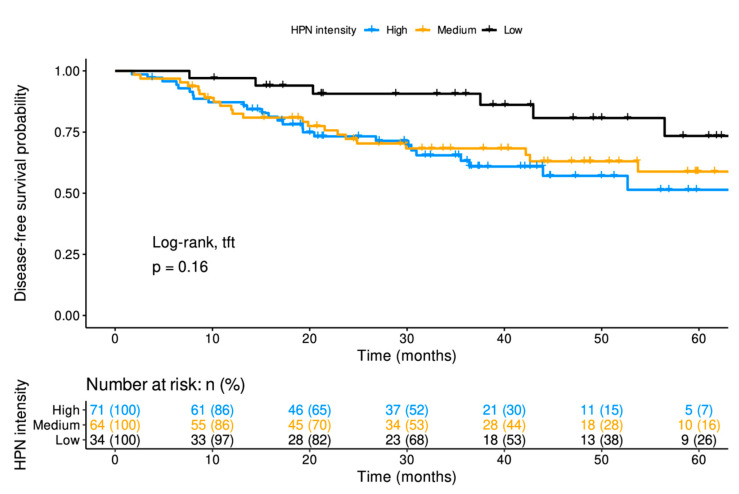
Disease-free survival of localized patients at diagnosis according to Hepsin staining. Disease-free survival is shown for the three groups of patients with different hepsin levels at diagnosis. Time on the X-axis corresponds with months of follow-up since diagnosis. The Log-rank test for trends gives information on the degree of significance of the differences between the three groups of patients. At the bottom, the patients who may suffer metastatic relapse after a given time from the start of follow-up are shown. *p*: *p*-value; tft: test for trends; HPN: Hepsin; n (%): number of patients (percentage).

**Figure 3 cancers-14-03106-f003:**
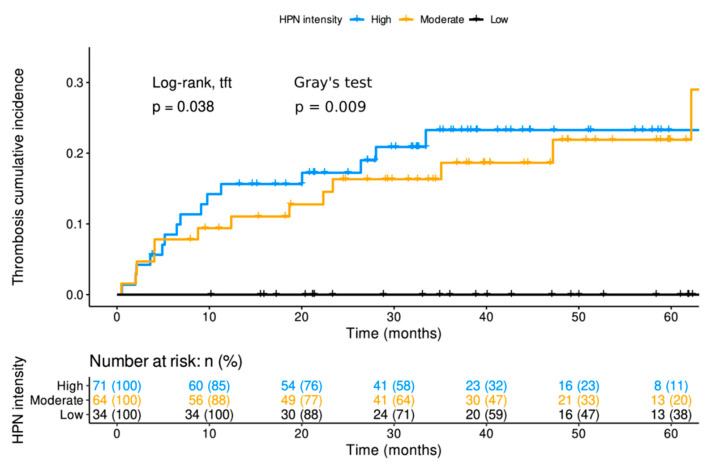
Cumulative incidence of thrombosis of localized patients at diagnosis according to Hepsin staining. The cumulative incidence of thrombosis is shown for the three groups of patients with different hepsin levels at diagnosis. The time on the X-axis corresponds with the months of follow-up since diagnosis. The Log-rank test for trends and the Gray’s test give information on the degree of significance of the differences between the three groups of patients. At the bottom, the patients who may suffer from thrombosis after a given time from the start of follow-up are shown. *p*: *p*-value; tft: test for trends; HPN: Hepsin; n (%): number of patients (percentage).

**Figure 4 cancers-14-03106-f004:**
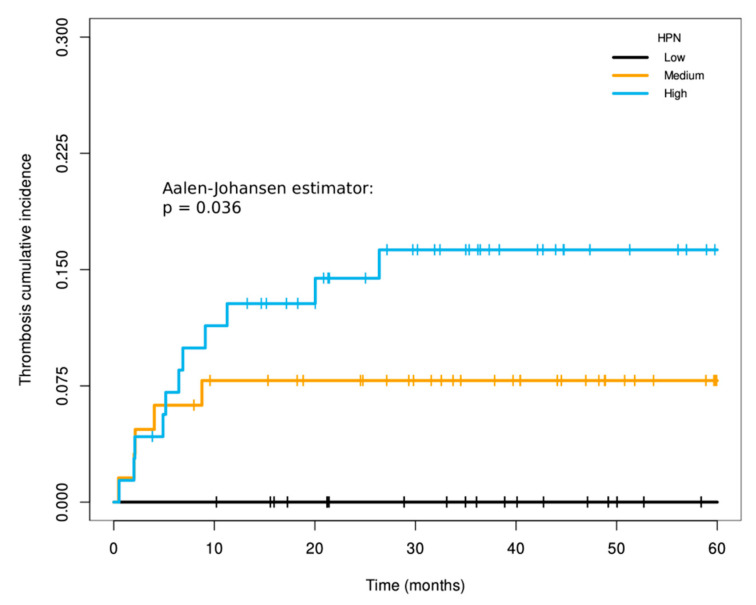
Cumulative incidence of thrombosis independent of relapse among localized patients according to Hepsin staining. Time on the X-axis corresponds with the months of follow-up since tumor diagnosis. The *p*-value from the Aalen–Johansen estimator gives information on the degree of significance of the differences between the three groups of patients. *p*: *p*-value of the Aalen–Johansen estimator; HPN: Hepsin.

**Figure 5 cancers-14-03106-f005:**
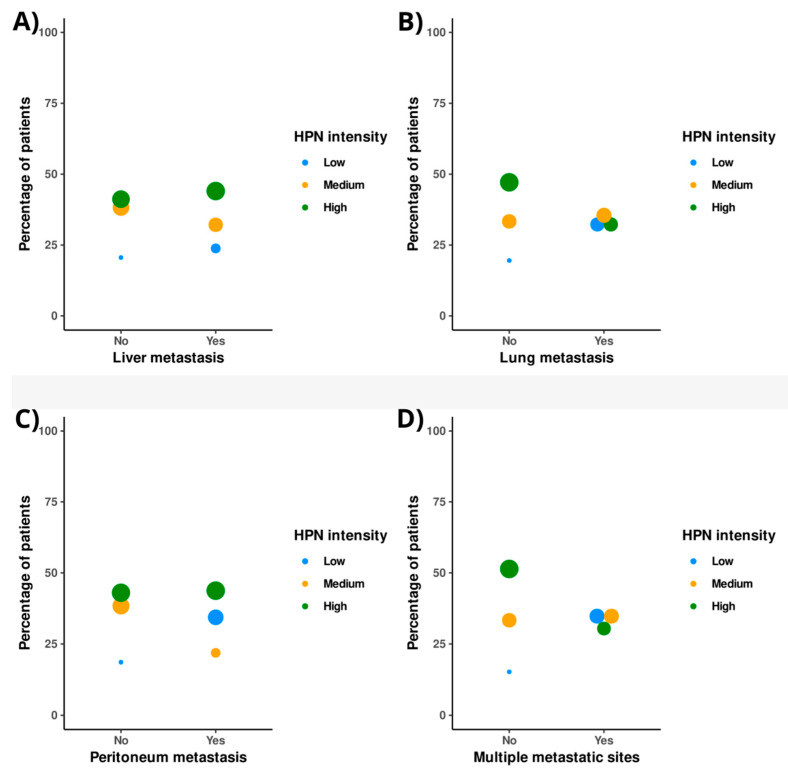
Distribution of hepsin staining intensity in patients with different metastatic involvement at diagnosis of cancer. (**A**) Liver metastasis. (**B**) Lung metastasis. (**C**) Peritoneal metastasis. (**D**) More than one metastatic location. HPN: Hepsin.

**Table 1 cancers-14-03106-t001:** Clinical and histopathological characteristics of patients from the study.

Clinical/Histopathological Features	Overall Cohort N ^1^ = 287	Localized Patients N = 169 (58.9%)	Metastatic Patients N = 118 (41.1%)
Median age at diagnosis (Range) (years)	66 (22–88)	64 (22–83)	68 (26–88)
Males: N (%)	187 (65.2)	109 (64.5)	78 (66.1)
Median follow-up (Range) (months)	35.1 (1.4–194)	41.2 (3.8–190.8)	24.5 (1.4–194)
ECOG ^2^ at diagnosis < 2: N (%)	237 (82.6)	152 (89.9)	85 (72)
Median survival (95% CI ^3^, LL ^4^-UL ^5^) (months)	64.5 (55.4–82.9)	93 (83.2–118.2)	24.7 (21.5–33.5)
Localized patients at diagnosis who undergo metastatic relapse during follow-up: N (%)	NA	54 (32)	NA
Patients with disease progression after first-line chemotherapy: N (%)	145 (50.5)	43 (25.4)	102 (86.4)
5-year cumulative incidence of thrombosis (%; 95% CI, LL-UL)	25.91; 20.49–32.43	18.24; 12.69–25.84	39.39; 29.05–51.85
Primary tumor			
Ascending colon: N (%)	83 (28.9)	48 (28.4)	35 (29.7)
Descending colon: N (%)	14 (4.9)	9 (5.3)	5 (4.2)
Transverse colon: N (%)	15 (5.2)	10 (5.9)	5 (4.2)
Sigmoid colon: N (%)	67 (23.3)	36 (21.3)	31 (26.3)
Rectal: N (%)	99 (34.5)	64 (37.9)	35 (29.7)
Multiple synchronous locations: N (%)	9 (3.1)	2 (1.2)	7 (5.9)
Localization of metastases at diagnosis			
Liver: N (%)	NA	NA	84 (71.2)
Lung: N (%)	NA	NA	31 (26.3)
Peritoneum: N (%)	NA	NA	32 (27.1)
Other affectations: N (%)	NA	NA	4 (3.4)
More than one metastatic site: N (%)	NA	NA	46 (39)
Histological grade at diagnosis			
Well differentiated: N (%)	127 (44.3)	90 (53.3)	37 (31.4)
Moderately differentiated: N (%)	85 (29.6)	53 (31.4)	32 (27.1)
Poorly differentiated: N (%)	28 (9.8)	10 (5.9)	18 (15.3)
Lymphovascular invasion at diagnosis: N (%)	123 (42.9)	76 (45)	47 (39.8)
Perineural invasion at diagnosis: N (%)	57 (19.9)	33 (19.5)	24 (20.3)
T-stage ^6^ > 2 at diagnosis: N (%)	191 (66.6)	127 (75.1)	85 (72)
N-stage ^7^ > 0 at diagnosis: N (%)	139 (48.4)	90 (53.3)	49 (41.5)
HPN staining intensity at diagnosis			
High: N (%)	122 (42.5)	71 (42)	51 (43.2)
Medium: N (%)	104 (36.2)	64 (37.9)	40 (33.9)
Low: N (%)	61 (21.3)	34 (20.1)	27 (22.9)

N ^1^: Number of patients; ECOG ^2^: Eastern Cooperative Oncology Group; CI ^3^: Confidence interval; LL ^4^: Lower limit; UL ^5^: Upper limit; T-stage ^6^: Size and extent of primary tumor; N-stage ^7^: Extent of tumor that had spread to nearby lymph nodes.

**Table 2 cancers-14-03106-t002:** Multivariate Cox regression model for disease-free survival in localized patients at diagnosis.

Multivariate Cox Regression for DFS ^1^						
Regressions	Coef ^2^	Exp (Coef) ^3^	LL ^5^ 95% CI ^4^ of Exp (Coef)	UL ^6^ 95% CI of Exp (Coef)	Se (Coef) ^7^	*p* ^8^
HPN ^9^ Medium	1.04	2.84	1.08	7.49	0.50	0.035 *
HPN High	1.20	3.30	1.30	8.41	0.48	0.012 *
HPN Low (reference)	-	-	-	-	-	-
Moderately differentiated histological grade	0.53	1.69	0.95	3.02	0.29	0.073
Poorly differentiated histological grade	0.97	2.65	0.84	8.29	0.58	0.095
Well-differentiated histological grade (reference)	-	-	-	-	-	-
Lymphovascular invasion	−0.45	0.63	0.34	1.17	0.31	0.146
Absent lymphovascular invasion (reference)	-	-	-	-	-	-
TNM stage III ^11^	−0.08	0.93	0.51	1.68	0.30	0.802
TNM stage II ^10^ (reference)	-	-	-	-	-	-

The risk of relapse is calculated according to Hepsin staining intensity, adjusting this calculation by adding different histopathological covariates to the model. DFS ^1^: disease-free survival; Coef ^2^: Cox regression coefficient; exp (coef) ^3^: hazard ratio; CI ^4^: confidence interval; LL ^5^: lower limit; UL ^6^: upper limit; se (coef) ^7^: standard error of Cox regression coefficient; *p* ^8^: *p*-value; HPN ^9^: Hepsin; TNM stage II ^10^: T-stage 3/4, N-stage 0; TNM stage III ^11^: T-stage 1/2/3/4, N-stage 1/2; *: significant *p*-value.

**Table 3 cancers-14-03106-t003:** Location of thrombosis depending on the tumor stage.

	Overall, N (%) ^1^	Localized, N (%)	Metastatic, N (%)
Head and neck	4 (6.2)	2 (6.6)	2 (5.7)
Head and neck + PE ^2^	3 (4.6)	0 (0)	3 (8.6)
Catheter related	2 (3.1)	0 (0)	2 (5.7)
Catheter related + PE	1 (1.5)	1 (3.3)	0 (0)
Splanchnic	9 (13.8)	6 (20)	3 (8.6)
Splanchnic + PE	3 (4.6)	0 (0)	3 (8.6)
Femoral	10 (15.4)	4 (13.3)	6 (17.1)
Femoral + PE	2 (3.1)	1 (3.3)	1 (2.9)
Calf vein ^5^	2 (3.1)	1 (3.3)	1 (2.9)
Calf vein + PE	1 (1.5)	0 (0)	1 (2.9)
Lower extremity, NOS	5 (7.7)	5 (16.7)	0 (0)
DVT ^3^, NOS ^4^	1 (1.5)	1 (3.3)	0 (0)
PE, NOS	22 (33.8)	9 (30)	13 (3.9)
Total	65 (100)	30 (100)	35 (100)

All of them were deep venous thromboses, except for four of them, which were arterial thromboses (localized: x2 splanchnic; metastatic: x1 splanchnic, x1 femoral). N (%) ^1^: number of patients (percentage); PE ^2^: pulmonary embolism; DVT ^3^: deep venous thrombosis; NOS ^4^: not otherwise specified. Note: calf vein ^5^ includes: anterior tibial/posterior tibial/fibular veins.

**Table 4 cancers-14-03106-t004:** Multivariate Fine and Gray regression model for the cumulative incidence of thrombosis in localized patients at diagnosis.

Multivariate Fine and Gray Regression for Cumulative Incidence of Thrombosis						
Regressions	Coef ^1^	Exp (Coef) ^2^	LL ^4^ 95% CI ^3^ of Exp (Coef)	UL ^5^ 95% CI of Exp (Coef)	Se (Coef) ^6^	*p* ^7^
HPN ^8^ Medium	2.04	7.71	1.06	55.92	1.01	0.043 *
HPN High	2.20	9.02	1.27	63.80	0.998	0.028 *
HPN Low (reference)	-	-	-	-	-	-
Moderately differentiated histological grade	0.10	1.11	0.54	2.29	0.37	0.780
Poorly differentiated histological grade	0.26	0.77	0.08	7.18	1.14	0.820
Well-differentiated histological grade (reference)	-	-	-	-	-	-
Lymphovascular invasion	0.37	1.45	0.68	3.07	0.38	0.330
Absent lymphovascular invasion (reference)	-	-	-	-	-	-
TNM stage III ^10^	−0.56	0.57	0.28	1.17	0.36	0.130
TNM stage II ^9^ (reference)	-	-	-	-	-	-

The risk of thrombosis is calculated according to Hepsin staining intensity, adjusting this calculation by adding different histopathological covariates to the model. Coef ^1^: Cox regression coefficient; exp (coef) ^2^: hazard ratio; CI ^3^: confidence interval; LL ^4^: lower limit; UL ^5^: upper limit; se (coef) ^6^: standard error of Cox regression coefficient; *p* ^7^: *p*-value; HPN ^8^: Hepsin; TNM stage II ^9^: T-stage 3/4, N-stage 0; TNM stage III ^10^: T-stage 1/2/3/4, N-stage 1/2; *: significant *p*-value.

## Data Availability

The original data are available upon request from the authors.

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
