# Peer review of "Implication of Hepsin from Primary Tumor in the Prognosis of Colorectal Cancer Patients"

_cancers, 2022, doi:10.3390/cancers14133106_

Round 1

Reviewer 1 Report

The limitations of this study:

1. Too small number of patients in the study group (only 287 patients with CRC),

2. Thrombosis very often accompanies cancer,

3. A single center study was conducted.

Author Response

We want to thank the comments of the reviewer. We have incorporated the limitations found by the reviewer in the main manuscript:

Among the limitations of this study, the first is the unique origin of the recruited patients, since they all belong to the same geographical area (Murcia Region, Spain), so the extrapolation of these results to larger cohorts from other distant geographical areas must be confirmed. The second limitation refers to the results regarding HPN as a thrombotic biomarker. Although the differences in the cumulative incidence of thrombosis are significant according to the expression of this protein in the primary tumor biopsy, thrombosis, which very often accompanies cancer, has a multifactorial etiology that involves not only genetic factors but also numerous environmental factors that could interfere with the usefulness of HPN as a biomarker.

Reviewer 2 Report

The manuscript entitled "Implication of Hepsin from primary tumor in the prognosis of colorectal cancer patients" of Zaragoza-Huesca et al., is well written, well organized and of great interest. I suggest this work for publication.

Author Response

We would like to thank the reviewer for the kind words. 

Reviewer 3 Report

Authors did well described the "Implication of Hepsin from primary tumor in the prognosis of colorectal cancer patients".

I do not feel any points that should be corrected.

Thus this paper could be published in current style.

Author Response

(The authors gave the same response as above.)
